

# A low-cost device for measuring local magnetic anomalies in volcanic terrain

Bertwin M. de Groot[1,2] and Lennart V. de Groot[1]

[1]Paleomagnetic laboratory *Fort Hoofddijk*, Utrecht University, Budapestlaan 17, 3584 CD Utrecht, Netherlands
[2]Technical & Analytical Support Earth Sciences, Utrecht University, Princetonlaan 8a, 3584 CB Utrecht, Netherlands

**Correspondence:** B.M. de Groot (b.m.degroot@uu.nl)

**Abstract.** Reconstructions of the past behavior of the geomagnetic field critically depend on the magnetic signal stored in extrusive igneous rocks. These rocks record the Earth's magnetic field when they cool and retain this magnetization on geological time scales. In rugged volcanic terrain, however, the magnetic signal arising from the underlying flows may influence the ambient magnetic field as recorded by newly formed flows on top. To measure these local anomalies in the Earth's magnetic field directly we developed a low-cost field magnetometer based on a flux gate sensor. To improve the accuracy of the obtained paleomagnetic vector and user friendliness of the device we combined this flux gate sensor with tilt and GPS sensors to rotate the measured magnetic vector to true North, East and down. The data acquisition is done using a ruggedized laptop and data are immediately available for first order interpretation. The first measurements done on Mt. Etna show local variations in the ambient magnetic field larger than expected, and illustrate both the accuracy (certainly $< 0.5°$ in paleomagnetic direction) and potential of our new device.

## 1 Introduction

The Earth's magnetic field has a pivotal role in the Earth Sciences, and has its applications in magnetostratigraphy, tectonics, and studies of the deep Earth. Furthermore, the Earth's magnetic field protects us against electromagnetically charged particles from the Sun that, if they would not be deflected by the Earth's magnetic field, would slowly strip away our atmosphere. An excess of such charged particles interferes with technological advancements such as wireless communication and satellites. Over the past centuries the Earth's magnetic field lost more than 20% of its strength, and regionally variations are even more dramatic (e.g. Pavón-Carrasco et al., 2014; Nilsson et al., 2014). To come to a thorough understanding of the behavior of the Earth's magnetic field it is paramount to have a record of the behavior of the Earth's magnetic field through (geologic) time and for different locations. The only recorder of the Earth's magnetic field that is available all over the globe and throughout geologic history are extrusive volcanic rocks, e.g. lava. Lava becomes magnetic when the iron-oxide bearing minerals cool trough their Curie temperature, and stores this magnetization even on geological timescales. By sampling many cooling units





with a known age in a volcanic edifice it is possible to reconstruct regional variations in the Earth's magnetic field for a certain region, while its resolution in time is determined by the availability of well-dated cooling units (e.g. de Groot et al., 2013a; Greve et al., 2017).

The methodologies of obtaining paleodirections and paleointensities from a single cooling unit have been tested by sampling
recent flows, e.g. flows that acquired their magnetization in a known magnetic field (e.g. Biggin et al., 2007; de Groot et al., 2013b). Especially the paleointensity proves to be very hard to reconstruct, often experiments that are deemed 'technically successful' produce under or over estimates of the known paleofield. Furthermore, paleodirections are sometimes hard to obtain reliably (e.g. Castro and Brown, 1987; Coe et al., 2014). Often, the reasons for these deviations are sought in rock-magnetic processes such as 'thermal alteration' or 'multidomain effects' that are known to hamper paleomagnetic experiments,
but it is also possible that these deviations from the expected intensities and directions actually arise from local magnetic anomalies caused by the magnetization of underlying lava flows. Local anomalies are known to cause deviations in magnetic compass readings in volcanic terrain, and they may therefore very well influence the magnetic field as recorded by lavas when they cool (Baag et al., 1995; Valet and Soler, 1999; Tanguy and Le Goff, 2004).

Here, we present a low-cost device that measures the ambient magnetic field at a selectable distance from the surface of
a lava flow to enable systematic mapping of local magnetic anomalies in volcanic terrain: the 'AnomalyMapper'. Its design revolves around a three-axis flux gate sensor that is mounted on an aluminium frame. To determine the declination, inclination, and intensity of the ambient magnetic field, we need to know the orientation of the flux gate sensor with respect to true (geographic) North, East, and down. To this end, there are two main hurdles to overcome: (1) it is impossible to align the flux gate sensor perfectly along the vertical while measuring in volcanic terrain, and (2) we cannot use a magnetic compass to
orient the flux gate to true North, as we are measuring local magnetic anomalies that interfere with compass readings. During normal operation it is possible to keep the AnomalyMapper upright within $\pm 3°$ of true vertical by using a bubble level. To also correct for the remaining deviation from vertical, we use an accelerometer (e.g. tilt sensor) that is fixed to the flux gate sensor to determine the orientation of the AnomalyMapper with respect to the direction of gravity; these measurements are used to rotate the flux gate measurements to true vertical. An intuitive way to avoid using a magnetic compass would be to use
a Sun compass, but this would render the AnomalyMapper useless when the sky is overcast. We therefore use a scope to orient the AnomalyMapper to a fixed reference point on the ground with a known (GPS) location. By logging the position of the AnomalyMapper for each measurement with a highly accurate GPS sensor we can determine the bearing of the measurement location to the reference point, and hence rotate the measurements to true North and East. This experimental design yields highly accurate magnetic measurements, while the measurements can be done quickly in the field.

To test the performance of the AnomalyMapper, we mapped local magnetic anomalies nearby and on top of a block of lava from the 2002-flow of Mt. Etna (Sicily, Italy) at three distances above ground. Furthermore, we assess the performance of the correction based on the tilt sensor to rotate the flux gate measurements to true vertical during normal operation, and under rather extreme circumstances in which the AnomalyMapper was held under angles up to $25°$ from true vertical.

Our AnomalyMapper is a low-cost device, and many parts are likely to be readily available in paleomagnetic laboratories.
Apart from the flux gate sensor that is commercially available for ~€2000, the set-up totals <€1500, including a ruggedized



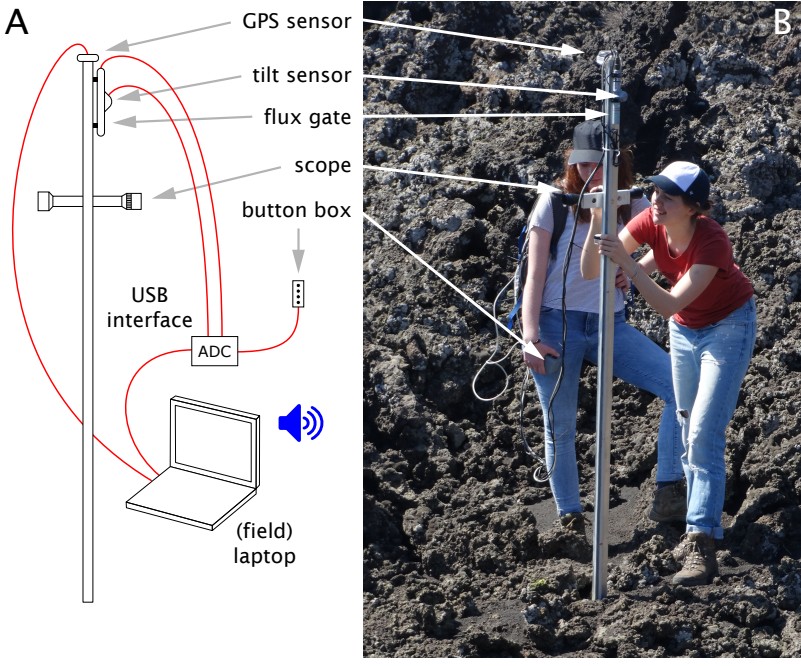

**Figure 1.** The AnomalyMapper. The AnomalyMapper consists of a GPS sensor, a tilt sensor, a flux gate sensor and a scope mounted on an aluminium frame, and is operated by pressing buttons on a button box (A). The electronic components are connected (red lines) to a ruggedized laptop through USB interfaces, the laptop has audible feedback to the users. The AnomalyMapper is easily operated by two people in the field (B), while the USB interface and laptop are carried in a small backpack.

laptop suitable for use in the field, tilt sensor, interfaces between analog sensors and the laptop, and all other hardware necessary to build the instrument.

## 2   Physical description

The backbone of the AnomalyMapper is a rectangular aluminium tube with dimensions $40 \times 40 \times 2000$ mm (Fig. 1A). An

5   aluminium I-profile was glued along its entire length, allowing the flux gate sensor to slide from top to bottom using aluminium mounts (in which the stainless steel bolts were changed by brass ones). The tilt sensor was glued to the flux gate sensor to ensure assessing the actual orientation of the flux gate sensor with respect to gravity. The GPS sensor was mounted at the top of the frame so that it is not obscured while the AnomalyMapper is in use. The scope (Lensolux 3-9×32, with cross-hair) was fixed onto an aluminium profile and bolted to the frame using brass nuts, washers, and bolts. About halfway along the frame a

10   two-dimensional bubble level was fixed to the frame using a small piece of aluminium profile (Fig. 1A).



## 2.1 GPS sensor

The GPS sensor is a vital part of the AnomalyMapper as the rotation towards true North and East depends on the accuracy of its known position. Here we use a commercially available Navilock 6004P MD6, based on an u-blox NEO-6P chipset. It has a horizontal accuracy <1 m, and a vertical accuracy of <2 m. This sensor directly connects with the ruggedized laptop through
its USB-interface.

## 2.2 Tilt sensor

The orientation of the AnomalyMapper with respect to gravity is measured by a three-axis accelerometer chip, here we choose an Analog devices ADXL335 chip on a SparkFun Breakout board (SEN-09269). The ADXL335 is a microelectromechanical system (MEMS) device with sensitivity corresponding to up to $0.1°$, albeit with less than ideal drift characteristics and offset
and sensitivity accuracy which require calibration and correction. Offset and sensitivity calibration values were established in the lab and drift correction values are calculated for each measurement session. This chip is powered from the laptop using a Seeed stepdown DC power converter based on an MP1584 chip from Monolithic Power Systems. The power supply voltage provided to the ADXL335 is measured simultaneously with each read-out, as the three analog accelerometer outputs are ratiometric to the power supply. Identical 1 Hz bandwidth resistor-capacitor (RC) low pass filters are used on all four channels
for increased noise reduction and accurate recording of the power supply voltage.

## 2.3 Flux gate sensor

We used a commercially produced flux gate sensor that was readily available in our paleomagnetic laboratory: the Bartington Mag-03MCES100 connected to a Bartington power supply and display unit. This flux gate has a dynamic range of 0 to 100 $\mu T$, well suited to measure the range of expected field intensities in volcanic terrain. It has a three axis analog output, so the
precision of the measurements is determined by the analog-digital (AD) converter used, we chose an 16 bit AD converter, leading to an effective precision of $\ll 25$ nT.

## 2.4 Interfacing & computer

The analog outputs from the tilt and flux gate sensors are connected to a USB DAQ device, here we used a Measurement Computing USB-1608G. In the field the main user interface is a handheld button box with four buttons. Each button gives a
label (1–4) to the measured data, so the four buttons can be used to measure at four different distances above ground, or to label repeated measurements at the same location. The button box is connected to digital inputs on the USB-1608G. A ruggedized Lenovo laptop runs the data collection software, requiring no user input in the field after initialization.

## 2.5 Software

The data collection software is written in LabVIEW 2017. The software continuously collects GPS data and, when the operator
presses a button, records data points for the fluxgate and tilt sensor. Data acquisition is simultaneous for all channels at a 10





ksps sampling rate for 1000 samples per analog channel per measurement. The mean values of these 100ms measurements are written into a .csv file along with the GPS position for that measurement location and the height of the measurement according to the button pressed. User feedback is an audible confirmation of a successful data point recording, or an audible warning when the GPS data is old or the fluxgate or tilt sensor data are outside expected bounds.

## 3   Data acquisition

The AnomalyMapper is easily operated by two people (Fig. 1B). The first aims the AnomalyMapper at the reference point by looking through the scope, while the second keeps the AnomalyMapper more or less upright by keeping an eye on the bubble level and acquires data by pushing the appropriate button on the button box. Aiming the AnomalyMapper at the reference target needs to be done with great care, as this orientation defines the measured declination of the magnetic field. If measurements are to be done at different heights above ground, it is easiest to use spray paint to indicate the measurement locations and follow the same section as many times as necessary instead of moving the flux gate along the frame of the AnomalyMapper multiple times per location.

## 4   Data processing & reference frames

The magnetic flux densities as measured by the flux gate must be rotated towards North, East, and down, to be informative on the full-vector of the Earth's magnetic field in a particular location. To this end four rotations are necessary: (1) align the tilt sensor measurements to the reference frame of the AnomalyMapper, (2) align the flux gate measurements to the reference frame of the AnomalyMapper, (3) rotate the $z$-axis of the flux gate measurements to vertical based on the tilt sensor measurements while preserving the orientation of its $x$-axis to the direction of the reference point, and (4) rotate the measured magnetic flux densities towards North, East, and down around the $z$-axis of the AnomalyMapper.

### 4.1   Aligning tilt sensor measurements to the AnomalyMapper's frame

The tilt sensor is attached to the flux gate such that gravity during normal (upright) use is distributed over the three axis of the sensor so that each axis performs optimally. To rotate the tilt sensor measurements to the coordinate system as defined by the frame of the AnomalyMapper ($x$ in the direction of the scope, $y$ to the right of the scope, and $z$ downwards along the rod, Fig. 2A), we define a rotation matrix. This rotation matrix is created with the readouts of the tilt sensor when the AnomalyMapper is successively oriented with its $x$ (top row), $y$ (middle row), and $z$ (bottom row) axes aligned with gravity. This yields the following rotation matrix for the tilt sensor, $\mathbf{G_f}$ in which the first character of the indices denotes which axis of the AnomalyMapper was aligned with gravity and the second character indicates the axis of the tilt sensor:

$$\mathbf{G_f} = \begin{bmatrix} g_{xx} & g_{xy} & g_{xz} \\ g_{yx} & g_{yy} & g_{yz} \\ g_{zx} & g_{zy} & g_{zz} \end{bmatrix}$$





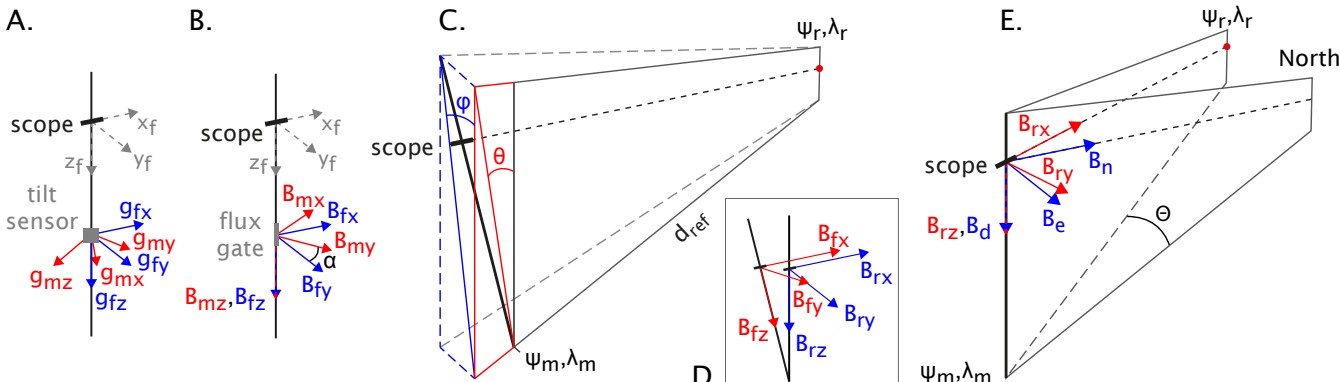

**Figure 2.** The different coordinate systems of the AnomalyMapper; in each panel the rotations are from the red to the blue coordinate systems. The axes of the tilt sensor ($g_{mx}$, $g_{my}$, $g_{mz}$) are rotated to the axes defined by the frame of the AnomalyMapper ($g_{fx}$, $g_{fy}$, $g_{fz}$) (A); and the axes of the flux gate ($B_{mx}$, $B_{my}$, $B_{mz}$) are rotated over angle $\alpha$ around the $z$-axes of the AnomalyMapper to $B_{fx}$, $B_{fy}$, $B_{fz}$ (B). Then the tilt sensor measurements are used to rotate the flux gate measurements in the system of the AnomalyMapper ($B_{fx}$, $B_{fy}$, $B_{fz}$) over angles $\phi$ and $\theta$ to the reference frame with $B_{rx}$ pointing in the direction of the reference point (red dot ($\Psi_r$, $\lambda_r$)), $B_{ry}$, and $B_{rz}$ being vertical (C, D). The last rotation is over angle $\Theta$ (the bearing from the measurement location ($\Psi_m$, $\lambda_m$) to the reference point) around the vertical axis to align $B_{rx}$ to the geographic North ($B_n$) and $B_{ry}$ to geographic East ($B_e$) (E).

The accuracy of the tilt sensor is affected by drift due to e.g. temperature differences, but the precision within a limited time span under constant conditions is very good (section 2.2). To correct for this absolute drift between different sites and the moment the rotation matrix $\mathbf{G_f}$ was determined, we average all measurements done at one site (usually $\gg$100 measurements) and assume that the average of these measurements represents the true vertical, i.e. has the same orientation as unit vector
5    [001]. The vector of the averaged tilt sensor measurements in its $x$, $y$, and $z$-directions should therefore be equal to the bottom row of $\mathbf{G_f}$. Differences between these two vectors arise from drift, and the measured data should be corrected for this before rotation matrix $\mathbf{G_f}$ can be used to align the tilt sensor data to the reference frame defined by the AnomalyMapper. Hence we define a correction vector ($\boldsymbol{\Delta g}$) as the difference between the averaged measurements in the $x$, $y$, and $z$-axes of the tilt sensor obtained at one site (and within a couple of hours), and the bottom row of rotation matrix $\mathbf{G_f}$:

10    $$\boldsymbol{\Delta g} = \begin{bmatrix} \overline{g_x} \\ \overline{g_y} \\ \overline{g_z} \end{bmatrix} - \begin{bmatrix} g_{zx} \\ g_{zy} \\ g_{zz} \end{bmatrix}$$

The tilt sensor data can now be rotated to the reference frame of the AnomalyMapper by correcting a measured tilt vector ($\boldsymbol{g_m}$) for drift, and multiplying it by the inverse of the rotation matrix:

$$\boldsymbol{g} = (\boldsymbol{g_m} - \boldsymbol{\Delta g}) \cdot \mathbf{G_f^{-1}}$$





## 4.2 Aligning flux gate measurements to the AnomalyMapper's frame

Although the flux gate was carefully aligned to the frame of the AnomalyMapper, a small, fortuitous, misalignment could not be avoided. It is possible to rotate the flux gate measurements to the reference frame defined by the AnomalyMapper using a similar rotation matrix as used for the tilt sensor, i.e. by creating a matrix with flux gate readouts while applying a magnetic

field successively in the three orthogonal axes. Since the $z$-axis of the flux gate is perfectly aligned with the $z$-axis of the AnomalyMapper due to its construction, however, we choose to carefully measure the deviation of the $x$ and $y$-axes of the flux gate with respect to the coordinate system of the AnomalyMapper ($\alpha$, Fig. 2B) with a protractor, and rotate the flux gates' measurements around its $z$-axis. This implies the use of the following rotation matrix $\mathbf{B_f}$; due to the alignment of the $x$, $y$, and $z$-axes with respect to the AnomalyMapper this rotation is in the negative direction (using an angle $-\alpha$):

$$\mathbf{B_f} = \begin{bmatrix} \cos(-\alpha) & -\sin(-\alpha) & 0 \\ \sin(-\alpha) & \cos(-\alpha) & 0 \\ 0 & 0 & 1 \end{bmatrix}$$

The measured flux gate data can now be rotated to the reference frame of the AnomalyMapper by multiplying a measured vector $\boldsymbol{B_m}$ by this rotation matrix:

$$\boldsymbol{B_f} = \boldsymbol{B_m} \cdot \mathbf{B_f}$$

## 4.3 Putting the flux gate measurements upright

While using the AnomalyMapper in the field great care is taken to position the stick upright; since the AnomalyMapper is handheld, however, deviations of up to $\pm 3°$ are common. With the data of the tilt sensor we can rotate the flux gate data to an upright reference frame with its $z$-axis vertical (Fig. 2C-D), but we have to be careful to preserve the orientation of the $x$-axis of the AnomalyMapper towards the reference point. To this end we apply two rotations, first around the $x$-axis of the new upright reference frame ($\phi$), and second around its $y$-axis ($\theta$). Due to the alignment of the $x$, $y$, and $z$-axes the rotation around

the $x$-axis is in the negative direction (using an angle $-\phi$), and the second in the positive direction (using an angle $+\theta$). The angles $\phi$ and $\theta$ are defined as:

$$\phi = \arctan\left[\frac{g_y}{g_z}\right] \quad \text{and} \quad \theta = \arctan\left[\frac{g_x}{g'_z}\right]$$

with $g_x$, $g_y$, and $g_z$ the tilt sensor data with respect to the frame of the AnomalyMapper (i.e. vector $\boldsymbol{g}$), and $g'_z$ the value of the $z$-axis of the tilt sensor data after the rotation around the $x$-axis of the new reference frame. The rotation matrices associated

with these rotations are:

$$\mathbf{R_1} = \begin{bmatrix} 1 & 0 & 0 \\ 0 & \cos(-\phi) & -\sin(-\phi) \\ 0 & \sin(-\phi) & \cos(-\phi) \end{bmatrix} \quad \text{and} \quad \mathbf{R_2} = \begin{bmatrix} \cos(\theta) & 0 & \sin(\theta) \\ 0 & 1 & 0 \\ -\sin(\theta) & 0 & \cos(\theta) \end{bmatrix}$$





To rotate the data to the reference system defined by the reference point and the vertical a multiplication of vector $\boldsymbol{B_f}$ with the two rotation matrices is sufficient:

$$\boldsymbol{B_r} = \boldsymbol{B_f} \cdot \mathbf{R_1} \cdot \mathbf{R_2}$$

### 4.4 Rotating flux gate measurements towards true North

The final step of the data processing is to rotate the flux gate data to true North using the locations of the reference point and the AnomalyMapper (Fig. 2E). To this end we have to determine the bearing ($\Theta$) from the measurement location (i.e. the location of the AnomalyMapper) to the reference point based on their GPS locations. Here we define the following: $\psi_m$, $\lambda_m$ the latitude and longitude of the measurement location, and $\psi_r$, $\lambda_r$ the latitude and longitude of the reference point. The bearing from the location of the measurement to the reference point with respect to true North is than given by:

$$\Theta = \arctan \left[ \frac{\sin(\lambda_r - \lambda_m) \cdot \cos(\psi_r)}{\cos(\psi_m) \cdot \sin(\psi_r) - \sin(\psi_m) \cdot \cos(\psi_r) \cdot \cos(\lambda_r - \lambda_m)} \right]$$

To rotate the vector $\boldsymbol{B_r}$ to geographic coordinates we define the following rotation matrix to rotate over an angle $-\Theta$ around the vertical axis:

$$\mathbf{R_3} = \begin{bmatrix} \cos(-\Theta) & -\sin(-\Theta) & 0 \\ \sin(-\Theta) & \cos(-\Theta) & 0 \\ 0 & 0 & 1 \end{bmatrix}$$

and multiply $\boldsymbol{B_r}$ with this matrix:

$$\boldsymbol{B} = \boldsymbol{B_r} \cdot \mathbf{R_3}$$

## 5   Experimental results

To assess the performance of the AnomalyMapper we mapped magnetic anomalies on a road cut in the 2002-flow of Mt. Etna (15.7957 °N, 15.0620 °E). The anomalies were measured at 5, 100, and 180 cm above ground, and we used a traffic sign approximately 200 m down the road as reference point. We measured a grid of $10 \times 11$ points in a rectangle of approximately

$20 \times 22$ m. The road and rock-face are roughly North-South. In each East-West line, the Easternmost four data points are on the road, then two or three points next to the rock-face, and the remaining four to five points are on top of the outcrop (Fig. 3A, B). The elevation was measured by the GPS sensor; although the accuracy of the GPS sensor (vertically <2 m) does not necessarily allow mapping the elevation of the outcrop properly, the main structures are produced very well (Fig. 3C).

The local variations in declination, inclination, and intensity are mapped in contour plots. The local anomalies are much

more prominent at 5 cm above ground, and become smoother at 100 and 180 cm above ground (Fig. 3). The magnetic field is more homogenous above the road, although the road is built on volcanic rock as well. Some features of the magnetic field correlate closely to the topography of the rock-face, e.g. the positive (37.79567 °N, 15.06193 °E) and negative (37.79563 °N,





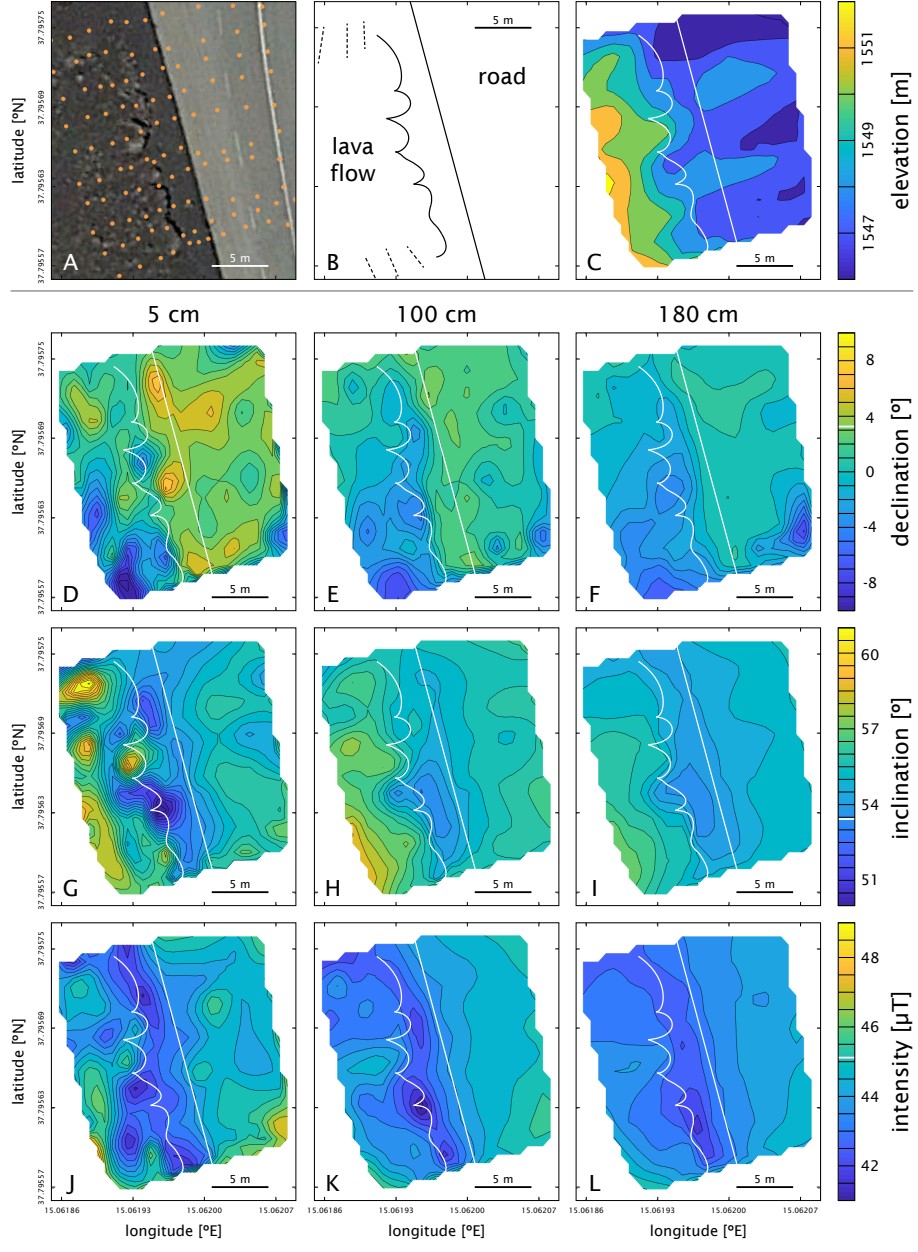

**Figure 3.** Experimental results as acquired on a road-cut in the 2002-flow of Mt. Etna. The GPS locations of the measurements are orange dots on a Google Earth image (A). The road, rock-face (wavy solid line) and shallower North and South slopes of the lava flow (dashed lines) are sketched (B), the rock-face and edge of the road are indicated in white in the other panels. The elevation was determined by the GPS sensor (C). The declinations (D-F), inclinations (G-I), and intensities (J-L) were measured at 5 (D, G, J), 100 (E, H, K), and 180 cm (F, I, L) above ground. For ease of comparison between panels with the same parameter the color schemes and contour lines are kept constant. The declination, inclination, and intensity as predicted by the IGRF (April 2018) are indicated as solid white lines in the scale bars.


15.06195 °E) anomalies in inclination at 5 cm above ground (Fig. 3G), and the low intensities at 100 cm above ground at 37.79563 °N, 15.06195 °E (Fig. 3K). Other anomalies at 5 cm above ground may also be due to the influence of loose boulders or rocks on top of the lava flow that were easily >30 cm in diameter (Fig. 3D, G, J).

## 6  Discussion

### 6.1  Experience in the field

The AnomalyMapper is portable, suitable for air travel and easy to use in the field. The acquisition of all data in Fig. 3 took less than 2.5 hours. Most parts of the AnomalyMapper are (or can be build) water resistant, so with proper precautions to protect the ruggedized laptop and the I/O device against rain it is possible to use the AnomalyMapper in most weather conditions.

The AnomalyMapper is most efficiently operated with two people: one aiming the AnomalyMapper at the reference point, while the other keeps the AnomalyMapper more or less upright by looking at the bubble level. When the magnetic anomalies are to be measured at more than one height above ground it is most efficient to use spray paint to mark the measurement locations and return to these points after adjusting the height of the flux gate.

### 6.2  Accuracy & performance

The tilt sensor is an important part of the design of the AnomalyMapper, as it enables accurate measurements when the AnomalyMapper is not exactly aligned with true vertical. To assess the performance of the tilt sensor we did 12 measurements in front of Paleomagnetic laboratory *Fort Hoofddijk* at Utrecht University (52.08808 °N, 5.17016 °E) and processed the data with and without the tilt sensor correction. The International Geomagnetic Reference Field at this location at the time of measurements is a declination of $-2.3°$, and an inclination of $66.8°$, with an intensity of $48.8\mu T$, although it must be noted that the ambient field at the measurement location may slightly deviate from these values. Since the intensity measurements are not affected by the position of the AnomalyMapper we can compare the averaged measured intensities with its one standard deviation $(48.1 \pm 0.03\mu T)$ directly to the reference field: the measured intensity is very close to the IGRF value, but slightly lower.

During normal operation the AnomalyMapper can be kept within $< 3°$ of true vertical using the bubble level (Fig. 4, LHS panels). Before tilt correction the declination (with its one standard deviation) is $-1.1 \pm 1.2°$, and the inclination $66.0 \pm 0.9°$. After tilt correction the declination and inclination become $-3.5 \pm 0.5°$, and $65.2 \pm 0.1°$, respectively. Both the declination and inclination are very close to the IGRF values, although the tilt sensor seems to correct the inclination in the wrong direction: the inclination before tilt correction is closer to the IGRF value than after the tilt correction. Due to local deviations in the magnetic field it must be emphasized that the IGRF does not necessarily represent the ambient magnetic field at the measurement location.

We then repeated the 12 measurements, but allowed the AnomalyMapper to deviate up to $25°$ from true vertical (Fig. 4, RHS panels). Before tilt correction this yielded an average declination of $-4.2 \pm 8.8°$, and $64.6 \pm 2.1°$ as inclination. After correction





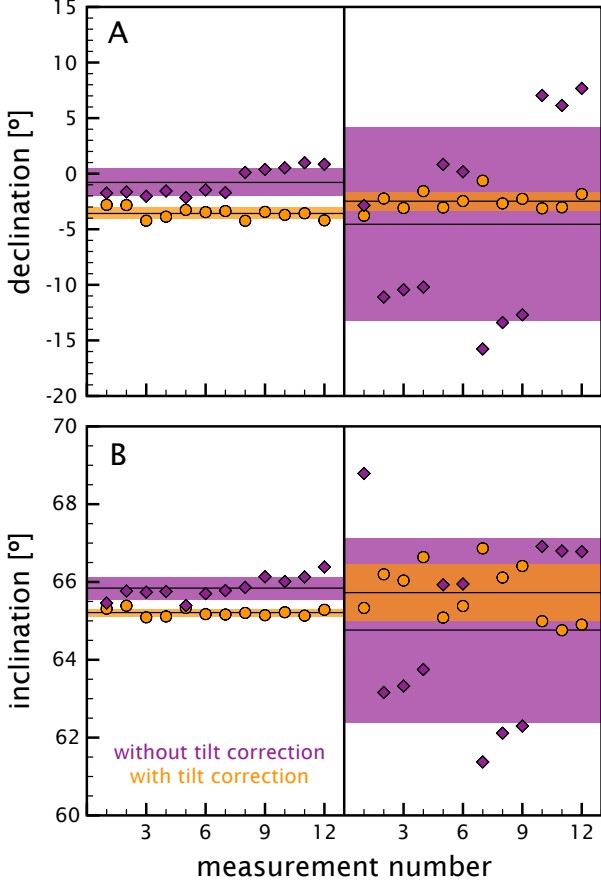

**Figure 4.** Comparison of the declination (A) and inclination (B) before and after tilt correction. We did 12 measurements while carefully positioning the AnomalyMapper upright using the bubble level (deviation form vertical $< 3°$, left hand side of both panels) and 12 measurements with the AnomalyMapper more or less upright (deviation form vertical up to $25°$, right hand side of both panels). Each measurement is plotted with (orange circles) and without (purple diamonds) tilt correction; the average of the groups of 12 measurements are given as horizontal black lines with their associated one standard deviation intervals as shading in the corresponding color.

using the tilt sensor data the declination and inclination became $-2.5 \pm 0.9°$, and $65.7 \pm 0.7°$, respectively. Here, the tilt sensor corrects the obtained declinations and inclinations to values closer to the IGRF reference field, and –more importantly– to values closer to those obtained by keeping the AnomalyMapper $< 3°$ of true vertical.

It is reassuring to note that the tilt sensor correction reduces the standard deviation associated with the declinations and inclinations dramatically; this implies that the measured values converge towards their mean after tilt correction. Moreover, the declinations and inclinations for the measurements done with the AnomalyMapper within $< 3°$ and with deviations up to $25°$, are pretty close, further testifying to the improvements in accuracy using the tilt sensor.



## 6.3 Drift correction of tilt sensor

The output of the tilt sensor suffers from long-term drift due to its mechanical nature, and from some temperature drift, both affecting sensitivity and offset. This means that the tilt sensor data cannot be used in absolute terms, but, since the drift is limited over the course of a couple of hours, we can use the mean value of all the data points in one measurement session to create

an assumed true vertical vector for that measurement session. The assumption here is that over the course of a measurement session the AnomalyMapper is on average held upright. The rotation matrix from the true vertical vector initially established in the lab to the assumed true vertical vector from the mean value for each session is applied to all data points in that session, thus providing individual long term drift correction for each measurement session.

## 6.4 Choosing the reference target

Choosing the proper reference target is paramount, it is important to choose a point that can be seen from all measurement points. The GPS sensor that determines the location of the AnomalyMapper has an accuracy of $< 1$ m. With a target at a distance of 200 m, the maximum deviation in GPS position in the least favorable direction leads to an error in the bearing between the measurement and reference point locations of $< 0.3°$. Choosing the target even further away, at i.e. 1 km, reduces this error to $< 0.06°$. In these calculations the GPS location of the reference point is considered accurate, as this location can

be measured multiple times to improve the accuracy of the GPS location, and can often easily be verified by satellite imagery.

## 7 Conclusions

The AnomalyMapper is an accurate, easy to use, and low-cost device to measure local magnetic anomalies in volcanic terrain. Considering the reproducibility of the measurements during normal operation, and choosing the reference target well, the AnomalyMapper is capable of determining declinations and inclinations with an accuracy of at least $< 0.5°$. Data acquisition

is quick: a grid of 110 points can be measured at three heights above ground within 2.5 hours. By making use of a reference point on the ground to align the coordinate system of the AnomalyMapper to true North, East, and down and a tilt sensor to rotate the flux gate measurements to true vertical the accuracy of the measurements is greatly improved. This experimental design also allows the AnomalyMapper to be used in all kinds of weather, except for very dense fog. The AnomalyMapper can be built for $<$ €1500 if a commercial flux gate sensor is at hand, otherwise the costs total at ~€3500 for the entire set-up.

*Data availability.* Both the raw data as measured by the AnomalyMapper and the processed data of the experiment on Mt. Etna are available as supplementary files to this manuscript

*Author contributions.* BMdG designed and built the instrument with help of LVdG. LVdG prepared the manuscript with contributions of BMdG.

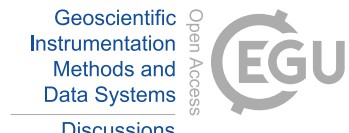

*Competing interests.* The authors declare no competing interests.

*Acknowledgements.* Wout Krijgsman, Maartje van den Biggelaar and Lynn Vogel helped acquiring the data on Mt. Etna presented in this manuscript; Lynn Vogel processed the data for which she is gratefully acknowledged. LVdG acknowledges NWO VENI grant 863.15.003.



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
