# Peer review of "A low-cost device for measuring local magnetic anomalies in volcanic terrain"

_Geoscientific Instrumentation, Methods and Data Systems, 2018_

## Short Comment (SC1) · 12 Dec 2018

The article describes a hand held device to measure magnetic field directions and intensities close to the ground. For that the authors combine standard components (magentic field sensor, inclinometer, gps, scope and bubble leveler) in a smart and easy way to obtain magnetic readings in a fixed earth reference frame. The article is well written and provides sufficient information about the construction and the data processing.

The weak point is the evaluation of the accuracy of the instrument. The author's claim "certainly <0.5° in paleomagnetic direction" is not backed up by statistical sound reference values. There are some factors that should be tested to show that the device accuracy is really valid under common field conditions.

1) Temperature coefficients of all components. This should be tested.

2) GPS accuracy. The authors assume <1m horizontally. The manufacturer's handbook states "Demonstrated under following conditions: 24 hours, stationary, first 600 seconds of data discarded". However waiting for 10 minutes at each measurement point is not very convenient in the field. Additionally the limited visibility of satellites due to topography can further affect the GPS accuracy. This can of course be overcome by using greater distances but should be tested and discussed in the paper.

3) The authors compare their measurements to the IGRF value which represents a global field model and can deviate locally. Hence it would be much better to compare the obtained field directions to in situ reference measurements with a proven instrument. Non-magnetic theodolites with attached single axis flux gate sensors are commonly used in geomagnetic observatories and can easily provide such measurements. Additionally the authors could use known field directions generated by helmholtz coil systems to demostrate the accuracy of their system for different field conditions without the need to travel to many different locations.

I recommend this article for publication once the authors provide a statistically solid demonstration of the accuracy of their instrument.

---

## Referee Comment (RC1) · Anonymous Referee #3 · 1 Apr 2019

The authors study the AnomalyMapper, that they describe like an accurate, easy to use, and low-cost device to measure local magnetic anomalies in volcanic terrain. For this hand held device, the accuracy of the determination of declinations, the time of data acquisition and the use under all kinds of weather are analyzed with very good results.

The mansucript is well organized and written. The methodology used and the results are very good described and supported. Therefore, I recommend the publication of this manuscript.

[Figure]

https://doi.org/10.5194/gi-2018-44, 2018.

---

## Referee Comment (RC2) · Anonymous Referee #4 · 8 Apr 2019

The paper is suitable to be published but for one thing. The authors give an opinion without references on a topic which has become lately of controversy:

line 15:

"... that, if they would not be deflected by the Earth's magnetic field, would slowly strip away our atmosphere."

The latest data from Mars and Venus missions have shown inconsistencies of the hypothesis that the lack of an intrinsic planetary magnetic (dipolar) field would result on the Solar wind blowing away eventually the atmosphere of a planet. Since the authors mention this without giving a reference, I suggest they add a reference or write

something of the form:

" could, allegedly, slowly strip away our atmosphere."

This does not affect the paper, per se, and it should not be a problem.

———————————————————

---

## Author Comment (AC1) · 25 Jun 2019

We thank the author for his/her positive comments and the recommendation to publish the manuscript as is.
* * *

---

## Author Comment (AC2) · 25 Jun 2019

Bertwin M. de Groot and Lennart V. de Groot

l.v.degroot@uu.nl

We rephrased the sentence on electromagnetically charged particles in the introduction of our paper to 'could strip away our atmosphere'. We thank the reviewer for his/her positive comments and recommendation on our manuscript.

---

## Author Comment (AC3) · 1 Jul 2019

This reviewer raises a number of valid points that we addressed in the main text as follows:

Ad 1) The temperature coefficients of the parts of the AnomalyMapper are important to consider. We therefore added a paragraph to the discussion (paragraph 6.3) devoted entirely to the thermal behavior of our device. It turns out that only the tilt sensor is (very) susceptible to changes in temperature. We therefore analyzed the thermal behavior of this chip against a superior alternative (the ADXL354) over the course of five days and a temperature range of 17.5 to 35.6 degrees C. We come to the

[Figure]

conclusion that the superior alternative of the chip should be used for future designs, but that by the drift correction in the data processing we already did (now paragraph 6.4), we can safely use our tilt sensor data.

Ad 2) The GPS indeed needs some time to get a proper satellite fix, but only when starting the device. We continuously read out the GPS sensor so the sensor keeps tracking satellites, and only 1 s is necessary to provide an accurate position of the AnomalyMapper. We added this to the paragraph on the GPS sensor, paragraph 2.1.

Ad 3) For the revision of this manuscript we obtained a proper measurement of the Earth's magnetic field at the location of the measurements using a horizontal surface with a precision of 0.03 degrees and flux gates. We now compare our measurements to this reference value that was obtained one year after the initial measurements. It is important to note that we cannot exclude that some natural or anthropogenic disturbances led to minor changes in the ambient field measured at Fort Hoofddijk in this one year. References to the IGRF in paragraph 6.2 were removed in favor of the actual, measured, reference value.

――――――――――――――――